# Mental and Physical Factors Influencing Wellbeing among South Korean Emergency Workers

**DOI:** 10.3390/ijerph18010070

**Published:** 2020-12-24

**Authors:** Mi Young Choi

**Affiliations:** Department of Emergency Medical Services, Sun Moon University, 70, Sunmoon-Ro 221 Beon-Gil, Tangjeong-Myeon, Asan-City 31459, Chungcheongnam-do, Korea; emtmy476@gmail.com or emtmy117@sunmoon.ac.kr

**Keywords:** emergency workers, stress, wellbeing, firefighters, paramedics

## Abstract

Emergency workers are frequently exposed to hazardous situations and such life patterns can influence their wellbeing. This study examined the relationships among South Korean emergency workers’ precedents and consequences of positive emotion, engagement, relationship, meaning, and achievement (PERMA), a wellbeing concept, and offered solutions. A total of 597 emergency workers in Daegu, South Korea, participated in a survey. This study measured post-traumatic stress disorder syndrome, burnout, depression, PERMA, quality of life, life satisfaction, and sleep quality to test the relationships. Results demonstrated that post-traumatic stress disorder syndrome and burnout predicted distracting sleep behavior and sleep health. Depression was significantly related to PERMA. The better the emergency workers’ PERMA was, the better their quality of life and life satisfaction were. PERMA significantly predicted sleep behavior, a portion of sleep quality. Depression had an indirect influence on quality of life mediated by PERMA. Post-traumatic stress disorder syndrome, burnout, and PERMA were significant predictors of low sleep health and sleep behavior. The results indicate that South Korean emergency workers struggle with depression and sleep quality. As the data were collected during the coronavirus disease 19 pandemic, individual efforts and relevant programs to improve South Korean emergency workers’ PERMA and sleep quality in a crisis are recommended. Possible solutions to improve the wellbeing of South Korean emergency workers are suggested.

## 1. Introduction

### 1.1. Background

As emergency workers (e.g., paramedics and firefighters) are frequently exposed to perilous situations resulting from car accidents, fires, and massive disasters, they tend to suffer from psychological disorder such as post-traumatic stress disorder syndrome (PTSD), burnout, and depression [1]. The emergency system in South Korea, an Asian country, is in transition. South Korea is still above the Organization for Economic Cooperation and Development (OECD) average in the annual number of deaths resulting from disasters [2]. South Korean experts in emergency medical services (EMS) argue that the country needs to improve its emergency medical system, decrease the heavy workload of emergency workers, and address the shortage of personnel [3]. For the last two decades, the South Korean EMS field has received substantial research attention in finding solutions for employees’ excessive workload and psychological health issues. Studies were conducted on the relationship between work stressors and quality of life [4], the influence of violent experiences and stressors on work satisfaction [5], and the association between violent experiences and work stress [6]. Despite such research attempts in South Korea, the studies dealt with individual factors for respective effects rather than compounding relationships. Little research has been conducted on explaining integrative mental and physical factors for the wellbeing of emergency workers in a South Korean context.

A positive psychology model offers a theoretical frame that considers multiple elements of wellbeing. Seligman recounts psychosocial wellbeing as positive emotion, engagement, relationship, meaning, and achievement (PERMA) [7]. The PERMA model regards five elements (positive emotion, engagement, relationship, meaning, and achievement) as motivators that account for wellbeing in life [7]. The PERMA model explains that the five elements can lead to a life of fulfillment and satisfaction [7]. Past research using the PERMA model found that high PERMA is negatively associated with PTSD, burnout, and depression [8]. The PERMA model was used to explain life satisfaction as a consequence among those with difficulties in life (e.g., individuals with disabilities) [9]. Psychological precedents (e.g., PTSD, burnout, and depression) and PERMA were significant factors associated with one another [8]. The PERMA model has also been employed to account for quality of life, life satisfaction, and sleep quality [9].

Grounded in the PERMA model, the current study examines the precedents (PTSD, burnout, and depression) and consequences (quality of life, life satisfaction, and sleep quality) of PERMA among South Korean emergency workers. This study investigates whether both inner and outer psychological and physical factors emergency workers experience during work account for wellbeing consequences. The results of the PERMA analysis may offer new explanatory mechanisms that can contribute to improving South Korean emergency workers’ lives and developing tailored programs. Moreover, the data collected during the coronavirus disease 19 (COVID-19) may suggest emergency workers’ wellbeing management strategies during a pandemic.

### 1.2. Precedents of Positive Emotion, Engagement, Relationship, Meaning, and Achievement (PERMA)

As a precedent to PERMA, PTSD is defined as a mental disorder stemming from experiencing terrifying events [10]. Emergency workers’ generativity, meaning in life, self-esteem, and social support are contributors to lower levels of PTSD [11]. Emergency workers with high PTSD experience low quality of life, excessive anxiety, and frights that negatively affect job implementation and family relationships [12]. The emergency workers who rescued victims in the aftermath of the 9–11 World Trade Center attacks later demonstrated high anxiety and low quality of life [13]. As seen in the research findings, PTSD likely predicts PERMA elements.

Burnout is a syndrome of physical, emotional, and mental states resulting from chronic workplace exposure that demands elevated emotion [14]. Emergency workers tend to experience burnout due to sustaining work stress [15]. As the level of burnout increases, quality of life decreases, and the health of emergency workers is affected negatively [4]. Given the research evidence, burnout would contribute to explaining PERMA.

Depression refers to a mental health state characterized by a persistently depressed mood, concern, feeling of loss, loss of self-esteem, or loss of interest in activities due to negative thoughts, one’s perceptions, environments, and future outcomes [16]. The PERMA elements are factors that negatively predict depression, stress, and anxiety [8]. Therefore, it is predicted that depression can be an influencing factor for PERMA.

### 1.3. Consequences of Positive Emotion, Engagement, Relationship, Meaning, and Achievement (PERMA)

The World Health Organization (WHO) defines quality of life as the perceptions about an individual’s status in life regarding the person’s goals, expectations, norms, and interests in the context of culture and value systems [17]. Emergency workers’ quality of life is influenced not only by health but also by PTSD, familial support, and health-related work restrictions [18].

Life satisfaction has been used as an indicator that tests the level of psychological wellbeing and health in many countries [19]. Butler and Kern argue that PERMA is a substantial component to account for life satisfaction [20]. Further, positive relational quality at home and work can diminish stress and facilitate quality of life and life satisfaction [21].

Sleep is one of the crucial parameters that affects a person’s quality of life and health [22]. Due to the work characteristics of frequent shifts, emergency workers experience sleep shortage, alcohol consumption, depression, anxiety, and distressing events [23]. Research evidence suggests that emergency workers’ wellbeing is likely significantly related to sleep quality.

Based on the review, this study tests whether a group of South Korean emergency workers’ precedents (PTSD, burnout, and depression) predict PERMA. Another set of predictions is to examine whether PERMA predicts consequences (quality of life, life satisfaction, and sleep quality). A proposed model is displayed in Figure 1.

## 2. Materials and Methods

### 2.1. Sample Data

This study used a purposive sampling method to collect data from emergency workers in South Korea. The goal of a purposive sample is to collect data from a representative sample of the population. This study attempted to accomplish the goal by applying expert knowledge to the population to select a non-random sample that represents a cross-section of the population [24]. The study’s population entailed all employees of the fire department stations in South Korea. This study used cross-sectional data collected from a survey of emergency workers. The sample respondents were emergency workers in the fire department stations in one of the major cities in South Korea. Data were collected from firefighters, paramedics, rescuers, administrators, and trainers in the fire department stations in Daegu, South Korea. In addition to firefighters, paramedics, and rescuers, administrators and trainers in the fire department stations were also included in the sample, because they deal with emergency-related tasks daily. Daegu was chosen for the following two reasons. First, the city is the third largest in South Korea and has a population of 2.5 million as of 2019. Second, as a metropolitan city, Daegu is the site of the subway arson disaster of 1995, the propane gas tank explosion of 1995, and the largest number of COVID-19 victims in February–March 2020.

A total of nine fire department stations (Regional Headquarter, Central, East, West, North, Suseong, Dalseo, Dalseong, and Kangseo) cover the districts in Daegu. Through the sampling frame of the emergency workers of the fire department stations in Daegu, the researcher sent a survey participation request to six stations (Headquarters, North, East, Central, Suseong, and Dalseo) and asked a representative of each station to distribute an online survey link to all emergency workers via email. Using purposive sampling, the six stations were chosen based on the researcher’s professional networks. In the requests, the researcher also asked the representatives to share the survey link with the other three stations (West, Dalseong, and Kanseo). As a result, 2595 firefighters, paramedics, and related employees (Regional Headquarter = 216, Central = 335, East = 313, West = 364, North = 302, Suseong = 318, Dalseo =232, Dalseong = 248, and Kangseo = 267) were reached for survey participation.

Upon approval from the Institutional Review Board (IRB) of the researcher’s university (Protocol Code: SM-202005-035-2), a survey was conducted from 1 July–8 September 2020, in which participants were provided an online survey link. The respondents who received a survey link via email visited the survey site and were asked to consent to participate in the survey. Only those who consented participated in the survey. The questionnaire was fully anonymous. No private information was collected. The questionnaire adapted the original scales developed in past studies and was translated into Korean. To verify accuracy and objectivity, a third researcher in a different field reviewed both the English and Korean versions. The third researcher verified the translation. Finally, the IRB confirmed the appropriateness of the questionnaire’s wording and approved translation.

### 2.2. Measurement

The current study used a total of seven variables as measurement instruments. An exploratory factor analysis (EFA) was conducted for the instruments’ validity and reliability, because the scales were reconstructed for the purpose of this study. Each component was determined based on eigenvalues over 1.0 and coefficients over 0.5 [25]. Component reliabilities were measured with Cronbach’s α for reliability with over 0.7 as an acceptable scale [26]. The EFA used the principal component with varimax rotation method. The EFA drew reliable components from eight items of PTSD, 12 items of burnout, eight items of PERMA, four items of quality of life, five items of life satisfaction, and 11 items of sleep quality. This study also conducted scale validity tests with the Kaiser–Meyer–Olkin (KMO) measure of sampling adequacy and Bartlett’s test of specificity. A single latent variable was created for each scale by summing and averaging. The latent variables were used for correlation and regression analysis in the preliminary phase. Factor coefficients, eigenvalues, variance explained, KMO measure of sampling adequacy, Bartlett’s test of specificity for model fit, and Cronbach’s Alpha were presented in Table 1. All factor Items are listed in Appendix A.

#### 2.2.1. Post-Traumatic Stress Disorder (PTSD)

PTSD measured the level of disturbing memories and thoughts due to stressful experiences of the past [27] and was rephrased for the current study’s context. Due to redundancies among the items in the other measurements of this study, only mutually exclusive eight items were selected (from 1 = Not at all to 5 = Very much so). An EFA yielded one factor for the eight items.

#### 2.2.2. Burnout

The items of burnout measured the degree to which the respondents felt emotional fatigue in varying forms [28] (from 1 = Not at all to 5 = Very much so). An EFA of 12 items produced unidimensionality with one factor. There were no item dropouts.

#### 2.2.3. Depression

Depression was assessed with the state of emotional anxiety and stress [29]. Four items with high coefficients verified in past research were selected to measure (from 1 = Do not apply to me to 5 = Applies to me all the time). An EFA found one dimension without any item dropouts.

#### 2.2.4. Positive Emotion, Engagement, Relationship, Meaning, and Achievement (PERMA)

As the wellbeing measure, a total of eight PERMA items were adapted for an EFA [30] (from 1 = Not at all to 5 = Always) and rephrased for the South Korean emergency worker context. After removing overlapping items with the other measurement scales, eight items were chosen for analysis. The eight items represented the PERMA elements, respectively. The scale assessed the levels of PERMA. By contrast with the anticipation of multiple factors, an EFA generated a single scale for the eight items.

#### 2.2.5. Quality of Life

The quality of life measure was operationally defined as the level of quality the respondents feel in the health, joy, and meaning of daily life [31] (from 1 = Not at all to 5 = Very much so). This study adapted the World Health Organization Quality of Life-brief (WHOQOL-BREF) scale and selected items with top four factor coefficients in past research [31]. The result of an EFA with four items produced unidimensionality.

#### 2.2.6. Life Satisfaction

The scale assessed the degree of satisfactory feelings in everyday life and used the satisfaction with life scale (SWLS) [32] (from 1 = Not at all to 5 = Very much so). Only five items with top factor coefficients in past research were adapted and analyzed in EFA [32]. The analysis yielded one factor without any dropouts.

#### 2.2.7. Sleep Quality

The 11 items of sleep quality measured health and behaviors during sleep [33] (from 1 = Not at all to 5 = Very much so). High scores indicated low sleep quality. An EFA yielded two components: sleep health (seven items) and sleep behavior (four items).

#### 2.2.8. Statistical Analysis

This study used Statistical Product and Service Solutions (SPSS) 26.0 for demographic characteristics, exploratory factor analysis, correlations, and regressions. A structural equation modeling test was implemented with Analysis of a Moment Structures (AMOS) 26.0.

## 3. Results

This study collected 597 responses representing a response rate of 23.01%. A G*Power test was conducted to draw a reliable number of respondents with effect size (.15), a Bonferroni-corrected α error probability of 0.05, and a 0.80 level of power (1-β) [34]. The result yielded a minimum sample size of 250 for valid results. The number of respondents met an acceptable sample size for analysis based on the G*Power test. The sample was composed of 524 males (87.8%) and 73 females (12.2%). The gender distribution is analogous to that of the national data of South Korean emergency workers. According to the national data in 2020, the ratio of male emergency workers to female was 9:1 [35]. The demographic information is described in Table 2. For a comparison of the variables, a one-way analysis of variance (ANOVA) analysis was used for the dependent variables by position. Results demonstrated that there were no significant differences in PTSD, burnout, depression, PERMA, quality of life, life satisfaction, sleep health, and sleep behavior by position. The results suggest that regardless of positions, the sampled emergency workers—including rescuers, paramedics, firefighters, administrators, and trainers—showed the identical levels on the dependent variables (see Table 3 for means of variables).

The current study conducted a structural equation modeling (SEM) test to verify whether latent variables are conceptually and significantly linked to subsequent variables in the lives of South Korean emergency workers. Correlation analysis was implemented as a preliminary test among the exogenous and endogenous variables (Table 3). The relationships of all variables turned out to be statistically significant. PTSD and PERMA were negatively associated with each other (*r* = −0.39, *p* < 0.001). Burnout (*r* = −0.46, *p* < 0.001) and depression (*r* = −0.49, *p* < 0.001) were also associated with PERMA negatively. PERMA was positively related to quality of life (*r* = 0.68, *p* < 0.001), life satisfaction (*r* = 0.78, *p* < 0.001), but negatively with sleep health (*r* = −0.40, *p* < 0.001), and sleep behavior (*r* = −0.33, *p* < 0.001).

As an additional preliminary analysis, this study conducted multiple regressions with the same independent and dependent variables. In predicting PERMA after controlling for demographics, burnout (β = −0.17, *p* < 0.05) and depression (β = −0.36, *p* < 0.001) were negatively associated with PERMA. The regression model explained 25.8% of the total variance (F = 22.73, *p* < 0.001). Another regression analysis with PTSD, burnout, depression, and PERMA as the independent variables and quality of life, life satisfaction, and sleep quality as the dependent variables was implemented. PERMA significantly predicted quality of life (β = 0.62, *p* < 0.001), life satisfaction (β = 0.69, *p* < 0.001), sleep health (β = −0.09, *p* < 0.01), and sleep behavior (β = −0.09, *p* < 0.05). PTSD predicted sleep health (β = 0.29, *p* < 0.001) and sleep behavior (β = 0.25, *p* < 0.001). Burnout predicted quality of life (β = −0.13, *p* < 0.05), life satisfaction (β = −0.29, *p* < 0.001), sleep health (β = 0.24, *p* < 0.001), and sleep behavior (β = 0.40, *p* < 0.01). Depression predicted sleep health (β = 0.18, *p* < 0.01). The regression models explained 49.4% of the total variance for quality of life (F = 57.14, *p* < 0.001), 64.8% for life satisfaction (F = 107.84, *p* < 0.001), 51.4% for sleep health (F = 61.97, *p* < 0.001), and 35.20% for sleep behavior (F = 31.90, *p* < 0.001).

The SEM test for the variables was conducted to yield an acceptable model with fit indices. Model fit indices included the goodness of fit index (GFI), comparative fit index (CFI), Tucker–Lewis index (TLI), and root mean square error of approximation (RMSEA). This study also used the degrees of freedom ratio over X^2^ to detect the acceptability of the model. The model is acceptable when the ratio is under 3.0 [36]. A model is considered acceptable when GFI is over 0.90, CFI over 0.90, TLI over 90, and RMSEA under 0.08.

The initial SEM test yielded an unacceptable model fit (X^2^ = 5337.72, df = 1264, *p* < 0.001, X^2^/df = 4.22, GFI = 0.711, CFI = 0.866, TLI = 0.860, RMSEA = 0.074). Modification indices suggested directional predictability from burnout to sleep health, from PTSD to sleep behavior, and from sleep behavior to sleep health in addition to the proposed relationships. Further, the model connected multiple residuals with each other to improve the model fit. The final model yielded model fit indices within acceptable ranges (X^2^ = 2120.30, df = 1160, *p* < 0.001, X^2^/df = 1.82, GFI = 0.900, CFI = 0.968, TLI = 0.964, RMSEA = 0.037) (Figure 2).

The final model showed that PTSD did not predict PERMA (β = 0.08, *p* > 0.05). Burnout was not significantly associated with PERMA either (β = −0.16, *p* > 0.05). Depression negatively predicted PERMA (β = −0.44, *p* < 0.001). PERMA positively and significantly predicted quality of life (β = 0.43, *p* < 0.001). The analysis also found a positive association between PERMA and life satisfaction (β = 0.74, *p* < 0.001). The relationship between PERMA and sleep health was not significant, whereas the association between PERMA and sleep behavior was significant and negative (β = −0.13, *p* < 0.01). Additional relationships demonstrate that PTSD contributed to disturbed sleep behavior (β = 0.52, *p* < 0.001). Burnout was a positive predictor of sleep health (β = 0.47, *p* < 0.001). Burnout also negatively predicted life satisfaction (β = −0.20, *p* < 0.001). Sleep behavior significantly affected sleep health (β = 0.51, *p* < 0.001).

In a post-hoc test examining direct, indirect, and total effects among the testing variables, a significant indirect relationship between depression and life satisfaction was found (β = −0.39, *p* < 0.01). There was no direct effect between depression and life satisfaction. The result means that an increase of one standard deviation in depression resulted in a negative increase in about four-tenths of a standard deviation in life satisfaction scores. The result indicates that PERMA plays a mediating role between depression and life satisfaction. The entire model explained 27% of the variance for PERMA, 80% for quality of life, 74% for life satisfaction, 75% for sleep health, and 37% for sleep behavior.

## 4. Discussion

The present study examined the precedents and consequences of PERMA, a wellbeing concept, among emergency workers in South Korea. Key findings suggest that a sample of South Korean emergency workers experience depression, which negatively influences PERMA. Moreover, PERMA is a significant indicator in predicting the respondents’ quality of life and life satisfaction. In particular, PTSD and burnout are deterrents of sleep quality. The results suggest that the respondents tend to grapple with depression, sleep health, and sleep behavior.

The result of the relationship between PTSD and PERMA is consistent with past research that found a significant relationship between PTSD and heavy workload but not PERMA [37]. The result indicates that the respondents may perceive the compounding influence of PTSD at the work level rather than solely at the personal level. Meanwhile, PTSD significantly predicted negative sleep behavior. The result is congruent with past studies [38,39], which found that sleep difficulty was deteriorated by PTSD and emphasized the importance of sleep quality for the improvement of PTSD. Sleep might be an issue of attention for South Korean emergency workers. The top concern of South Korean emergency workers was sleep disturbance, including insomnia and sleep apnea [40]. Experts on the treatment of emergency workers’ PTSD suggest working out, staying connected to support their fellow emergency workers, and getting educated about PTSD as solutions [41]. With these efforts, emergency workers may also improve sleep quality. Specific and practical therapies based on the results may improve the sleep quality of employees in the EMS field [42].

The non-significant relationship between burnout and PERMA is not consistent with past research [43]. It is possible that the relationship can be better explained with a moderator such as resilience [44]. Meanwhile, burnout negatively explained life satisfaction. Since burnout hinders a satisfying life, emergency personnel may seek some ideas of work-life balance [45]. As burnout creates a chronic motivational challenge, coping mechanisms such as trying to live an energetic life, developing positive aspects of life, and setting top priorities in daily lives can contribute to quality of life [46]. Those with PERMA in life could experience burnout less than those without PERMA [47]. Future research needs to elaborate on multiple factors including workload and emotional labor [48] in emergency workers’ work environments to account for burnout and PERMA.

The significant relationship between depression and PERMA implies that the depression of emergency workers can be a deterrent for life engagement, goal attainment, accountability, and happiness. In the SEM analysis, depression significantly predicted quality of life but became non-significant when PERMA was added as a mediator between depression and quality of life. The result suggests that PERMA may play a crucial role between emergency workers’ depression and quality of life [49]. As related research affirms [50], South Korean emergency workers’ efforts and some intervention programs and education to improve wellbeing may alleviate depression and enhance quality of life.

The positive relationship between PERMA and quality of life indicates that the respondents regard PERMA as an element for a joyful, meaningful, and healthy life with high morale [51]. Previous research that found quality of life facilitating resilience and health supports the current result [52]. Sustainable support for first responders is a significant enhancer for wellbeing and quality of life [12]. The positive relationship between PERMA and life satisfaction suggests emergency workers’ heightened attention to PERMA as a solution. Fire department stations’ concerted efforts at the organizational level to improve PERMA—which can facilitate life satisfaction, quality of life, and sleep quality—are recommended. The significant relationships among PTSD, burnout, and sleep quality imply that sleep health and sleep behavior are determined largely by the compounding factors, which suggests that increased attention is needed to facilitate better sleep quality [53]. Low PERMA could cause insomnia and shortened sleep duration [54]. The significant predictability of sleep behavior for sleep health implies that the respondents’ unhealthy sleep behavior may lead to hampered sleep wellbeing.

Through the results, the current study contributes to the EMS field both theoretically and practically. Theoretically, scant research attempts have comprehensively examined PTSD, burnout, depression, PERMA, quality of life, life satisfaction, and sleep quality among emergency workers. The current study is the first in South Korea to test the relationships between precedents and consequences of PERMA, a wellbeing concept, among emergency workers. An advanced theoretical model of PERMA may be developed from this research [7]. The theoretical framework used in this study can be applied to similar professionals, such as emotional laborers, hospital residents, and social welfare workers in both the private and public sectors. This study was conducted during the COVID-19 pandemic. Emergency workers were at the forefront of the crisis as first responders and coped with fear, anxiety, and uncertainty [55]. The significant relationships found in this study may reflect emergency workers’ work stress and consequences during a pandemic. The results demonstrate that South Korean emergency workers are distracted by sleep quality, which is affected by negative mental disorders such as PTSD and burnout amidst the COVID-19 pandemic.

In practical terms, the use of PERMA as a key factor delineating precedents and consequences of this study may offer ideas of program development (e.g., meaning of life, energetic life, top priorities in life, positive aspects of life, interaction with colleagues, healthy sleep, etc.) that can enhance the lives of emergency workers. The results of the present study suggest the factors (depression, PERMA, sleep quality, PTSD-sleep quality, and burnout-sleep quality) to emphasize further in program development. As such, the EMS field—including emergency workers in South Korea—may create tailored intervention programs for work-life balance. As emergency workers experience elevated mental disturbance levels during a pandemic [56], the consideration of PERMA and sleep quality may provide coping strategies for emergency workers in South Korea.

## 5. Limitations and Suggestions

This study has some conceptual and methodological limitations. First, the cross-sectional data from a city do not represent the entire population of South Korean emergency workers. Therefore, a national survey can be conducted in future research for representative results. Second, the structure of the sample had some limitations. The sample had only a small number of female emergency workers. Future research, therefore, could obtain a larger sample of female emergency worker participants by surveying medical emergency rooms or nurses. Police officers were not included in this emergency worker sample. Fire department stations and police department stations are two independent units in South Korea. The exclusion of police officers may limit the interpretation and the generalizability of the results. Furthermore, the current study included administration and training personnel in the sample. The small number of the administration and training personnel in the sample might not significantly affect the results of the analysis (*n* = 40, 6.7%). However, future research should focus on firefighters, paramedics, or police officers, respectively, to test each occupational group’s wellbeing. Third, the response rate (23.01%) was relatively low. Future research can obtain a higher response rate by promoting the benefits of participation. Fourth, with programs developed for emergency workers’ wellbeing, studies on the effects of the programs may be needed. Fifth, this study used some representing items of PERMA rather than the entire list of them. Had all five elements of PERMA as individual variables been used for analysis, more detailed relationships could have been yielded. Further, future research can apply the PERMA-V (vitality) model to strengthen the explanatory power of the model [57]. Finally, a comparative study of multiple countries’ emergency workers may provide cultural and international differences in the precedents and consequences of PERMA.

## 6. Conclusions

The current study provides a comprehensive theoretical model that tests precedents and consequences of PERMA, a wellbeing concept, applied to an emergency worker population. The development of PERMA facilitation and sleep quality improvement programs at both the individual and organizational levels can be practical suggestions for the lives of South Korean emergency workers. This study found that the sampled South Korean emergency workers’ PERMA, depression, PTSD, burnout, quality of life, life satisfaction, and sleep quality were significantly associated with each other. In particular, high depression lowered PERMA. PTSD and burnout negatively affected sleep quality. PERMA was a positive indicator for quality of life and life satisfaction. The importance of PERMA and sleep quality for both inner and outer psychological and physical factors was found in the lives of the respondents.

PERMA programs and comprehensive training may help not only emergency workers but also managers of fire department stations cope with job demands during their daily work. Policies and individual efforts to facilitate PERMA in general and sleep quality in particular may decrease stress and improve the work conditions of South Korean emergency workers.

## Figures and Tables

**Figure 1 ijerph-18-00070-f001:**
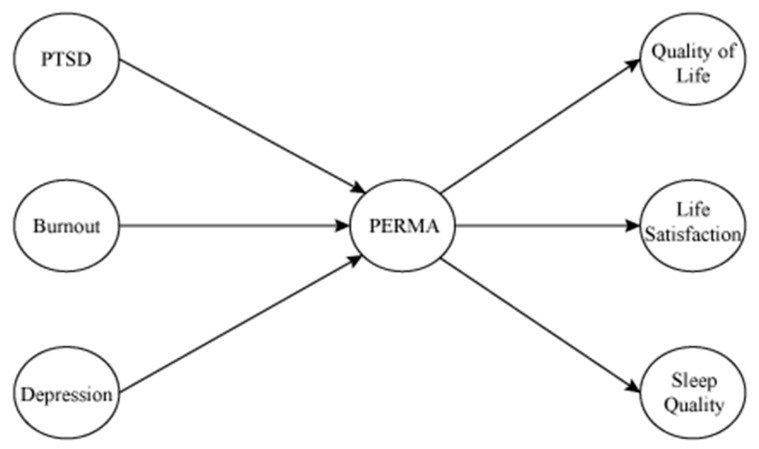
Proposed model of testing variables. Note. PTSD: post-traumatic stress disorder Syndrome; PERMA: positive emotion, engagement, relationship, meaning, and achievement.

**Figure 2 ijerph-18-00070-f002:**
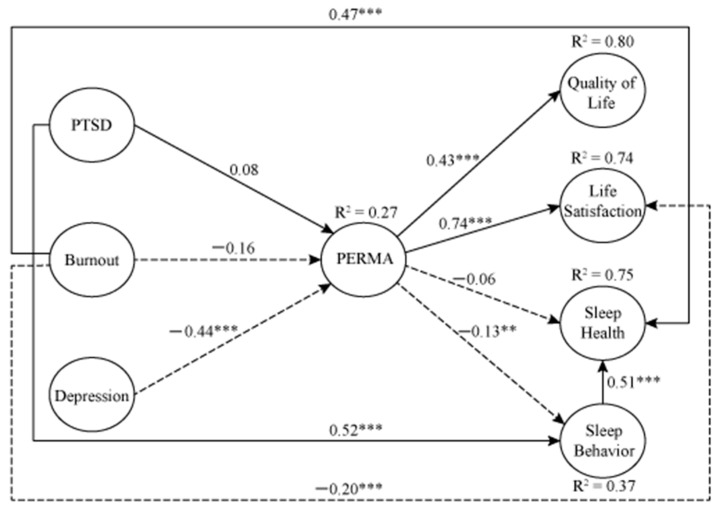
Results of testing variables. Note. PTSD: post-traumatic stress disorder syndrome; PERMA: positive emotion, engagement, relationship, meaning, and achievement. ** *p* < 0.01. *** *p* < 0.001.

**Table 1 ijerph-18-00070-t001:** Results of exploratory factor analysis (*N* = 597).

Item Number	PTSD	Burnout	Depression	PERMA	Quality of Life	Life Satisfaction	Sleep Health	Sleep Behavior
1	0.878	0.842	0.904	0.864	0.703	0.920		0.806
2	0.876	0.856	0.900	0.915	0.767	0.936		0.836
3	0.887	0.828	0.884	0.908	0.848	0.921		0.710
4	0.897	0.827	0.901	0.894	0.852	0.875	0.730	
5	0.849	0.838		0.891		0.891	0.638	
6	0.831	0.835		0.878			0.717	
7	0.851	0.814		0.873			0.642	
8	0.857	0.806		0.910			0.736	
9		0.853					0.640	
10		0.851					0.669	
11		0.845						0.681
12		0.816						
Eigenvalue	5.99	8.35	3.21	6.36	2.52	4.12	5.64	1.08
Variance Explained	74.99	69.62	80.47	79.53	63.14	82.56	51.29	9.87
KMO	0.933	0.953	0.855	0.949	0.647	0.897	0.906	0.906
Bartlett’s test (X^2^)	4631.63	6628.01	1715.37	5207.48	1131.42	2926.08	3338.89	3338.89
Degrees of Freedom	28	66	6	28	6	10	55	55
*p* value	0.000	0.000	0.000	0.000	0.000	0.000	0.000	0.000
Cronbach’s Alpha	0.95	0.96	0.92	0.96	0.79	0.94	0.87	0.83

Note. Items in number order are listed in Appendix A. PTSD: post-traumatic stress disorder syndrome; PERMA: positive emotion, engagement, relationship, meaning, and achievement. KMO: Kaiser–Meyer–Olkin measure of sampling adequacy.

**Table 2 ijerph-18-00070-t002:** Descriptive statistics of the sample (*N* = 597).

Variable	*N*	%
Gender		
Male	524	87.8
Female	73	12.2
Age		
20s	84	14.1
30s	360	60.3
40s	109	18.3
50s	42	7.0
Over 60	2	0.3
Education		
Under Middle School	12	2.0
High School	0	0.0
2-Year College	423	70.9
4-Year College	157	26.3
Graduate School	5	0.8
Position		
Rescuers	27	4.5
Paramedics	443	74.2
Firefighters	87	14.6
Administration	36	6.0
Training	4	0.7
Annual Income ($)		
Under 20,000	0	0.0
20,001–30,000	0	0.0
30,001–40,000	140	23.5
40,001–50,000	256	42.9
50,001–60,000	135	22.6
60,001–70,000	54	9.0
Over 70,000	12	2.0

**Table 3 ijerph-18-00070-t003:** Correlations among variables (*N* = 597).

	1	2	3	4	5	6	7	8
1. PTSD								
2. Burnout	0.79 ***							
3. Depression	0.76 ***	0.85 ***						
4. PERMA	−0.39 ***	−0.46 ***	−0.49 ***					
5. Quality of Life	−0.36 ***	−0.42 ***	−0.42 ***	0.68 ***				
6. Life Satisfaction	−0.41 ***	−0.51 ***	−0.49 ***	0.78 ***	0.67 ***			
7. Sleep Health	0.65 ***	0.66 ***	0.65 ***	−0.40 ***	−0.37 ***	−0.40 ***		
8. Sleep Behavior	0.53 ***	0.56 ***	0.49 ***	−0.33 ***	−0.35 ***	−0.41 ***	0.69 ***	
Mean	1.86	1.92	1.65	3.73	3.62	3.51	1.98	2.61
Standard Deviation	0.95	0.88	0.79	0.86	0.86	0.92	0.78	0.99
Skewness	1.07	0.84	1.28	−0.68	−0.27	−0.25	0.62	0.21
Kurtosis	0.40	0.06	1.39	0.79	0.04	−0.26	0.02	−0.54

Note. PTSD: post-traumatic stress disorder syndrome; PERMA: positive emotion, engagement, relationship, meaning, and achievement. *** *p* < 0.001.

## Data Availability

The data presented in this study are available on request from the corresponding author. The data are not publicly available due to the author’s possession on a password-protected computer.

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
