# Peer review of "Mental and Physical Factors Influencing Wellbeing among South Korean Emergency Workers"

_ijerph, 2020, doi:10.3390/ijerph18010070_

Round 1

Reviewer 1 Report

The author describes the results of a survey designed to assess diverse mental parameters in paramedics and firefighters working in the corean town of Daegu.

Major comments:
1. This study lacks a comparator
2. The introduction should focus on what is already known of the target population, and elements regarding other populations (first responders for example) should be moved to the discussion.
3. The use of PERMA within the text is often unclear. PERMA is a concept more than a score, and has therefore been adapted as PERMA-Profiler, to which the author refers (ref #36 - Umcu E et al, Psychometric validation of the PERMA-profiler as a well-being measure for student veterans, 2019). However the validated PERMA-Profiler is not used as such in this study (in PERMA profiler, there are 15 PERMA items and 8 filler items, which arerated on an 11-point scale ranging from 0 (never/not at all) to 10 (always/completely)). The author should make clearer whether the custom PERMA score or the PERMA concept are referred to.
4. There are results in the methods section ("The current study collected 597 responses representing a response rate of 23.01%. 129 The number of respondents met an acceptable sample size for analysis based on the G*Power test. The sample was composed of 524 males (87.8%) and 73 females (12.2%).") All results should be in the results section of the manuscript.
5. The response rate is low (< 25%), and there are no policemen in the "emergency workers" sample. This limits the interpretation and the generalizability of the results. Inclusion and exclusion criteria are not stated, which is problematic as including "Administration" and "Training" personnel in the analysis is debatable.

Minor comments:
6. South Korean experts in 31 emergency medical services (EMS) argue that the country needs to improve its EMS. In this sentence, the last EMS means "emergency medical system" rather than "emergency medical services" and the same abreviation should not be used.
7. "The PERMA model is also used to explain quality of life, 45 life satisfaction, and sleep quality as consequences among those with difficulties in life (e.g., 46 individuals with disabilities and emotional laborers)" - the reference is only about "Assessing College Life Adjustment of Students With Disabilities". The sentence should be updated to be more consistent with the reference cited (or a relevant reference could be added).
8. Between lines 56 and 64, there is no real "flow" and the sentences do not seem to be linked together
9. Some references are incorrectly formatted ("a .80 level of power (1-*beta*) (Cohen, 1988).")

Author Response

Major comments:
1. This study lacks a comparator

RESPONSE: Thank you for your comment Reviewer. To accommodate your point, I added a comparison of the dependent variables by position (rescuer, firefighter, paramedics, etc.). I also mentioned an international comparison of multiple countries’ emergency workers for future research in the limitation section.

In lines, 215-221,

“For the comparison purpose of the variables, a One-Way Analysis of Variance (ANOVA) analysis was used for the dependent variables by position. Results demonstrated that there were no significant differences in PTSD, burnout, depression, PERMA, quality of life, life satisfaction, sleep health, and sleep behavior by position. The results suggest that regardless of positions, the sampled emergency workers – including rescuers, paramedics, firefighters, administrators, and trainers – showed the identical levels on the dependent variables (see Table 3 for Means of Variables).”

In lines 378-380,

“Finally, a comparative study of multiple countries’ emergency workers may provide cultural and international differences in the precedents and consequences of PERMA.”

  1. The introduction should focus on what is already known of the target population, and elements regarding other populations (first responders for example) should be moved to the discussion.

RESPONSE: I agree with you. I revised the section significantly following your suggestion. I moved the studies on other populations to the discussion section. I targeted the emergency worker population only in the literature review section. I also discussed conceptual importance in the EMS context in the introduction section.

In lines 52-64,

“The PERMA model was used to explain life satisfaction as a consequence among those with difficulties in life (e.g., individuals with disabilities) [9]. Psychological precedents (e.g., PTSD, burnout, and depression) and PERMA were significant factors to be associated with one another [8]. The PERMA model has also been employed to account for quality of life, life satisfaction, and sleep quality [9].

Grounded in the PERMA model, the current study examines the precedents (PTSD, burnout, and depression) and consequences (quality of life, life satisfaction, and sleep quality) of PERMA among South Korean emergency workers. This study investigates whether both inner and outer psychological and physical factors emergency workers experience during work account for wellbeing consequences. The results of the PERMA analysis may offer new explanatory mechanisms that can contribute to improving South Korean emergency workers’ lives and developing tailored programs. Moreover, the data collected during the coronavirus disease 19 (COVID-19) may suggest emergency workers’ wellbeing management strategies during a pandemic.”

  1. The use of PERMA within the text is often unclear. PERMA is a concept more than a score, and has therefore been adapted as PERMA-Profiler, to which the author refers (ref #36 - Umcu E et al, Psychometric validation of the PERMA-profiler as a well-being measure for student veterans, 2019). However, the validated PERMA-Profiler is not used as such in this study (in PERMA profiler, there are 15 PERMA items and 8 filler items, which are rated on an 11-point scale ranging from 0 (never/not at all) to 10 (always/completely)). The author should make clearer whether the custom PERMA score or the PERMA concept are referred to.

RESPONSE: Thanks for your point. I used the original PERMA-Profiler for this study when building an argument and a questionnaire. Then I realized that there were redundant items overlapped with the other scales (e.g., life satisfaction and quality of life). To avoid conceptual and operational multicollinearity, I removed the redundant items from the PERMA scale. For response option consistency throughout the measurement, I used the same format of a 5-point scale in all measurement instruments for consistency. Even though I shortened the PERMA scale, I anticipated an EFA would produce multiple factors. However, the result was unidimensional. I fully agree with the representativeness issue of the PERMA scale in this study. In my future study, I will use the full list of the PERMA-Profiler to verify the validity of the scale. As an apparent study limitation, I mentioned it in the limitation section. Thank you for your understanding and please let me know what I can do to better present the results.

In lines 178-183,

“As the wellbeing measure, a total of eight PERMA items were adapted for an EFA [31] (from 1 = Not at all to 5 = Always) and rephrased for the South Korean emergency worker context. After removing overlapping items with the other measurement scales, eight items were chosen for analysis. The eight items represented the PERMA elements, respectively. The scale assessed the levels of PERMA. Different from the anticipation of multiple factors, an EFA generated a single scale for the eight items.”

In lines 375-378,

“Fifth, this study used some representing items of PERMA rather than the entire list of them. Had all five elements of PERMA as individual variables been used for analysis, more detailed relationships could have been yielded. Further, future research can apply the PERMA-V (vitality) model to strengthen the explanatory power of the model [58].”

  1. There are results in the methods section ("The current study collected 597 responses representing a response rate of 23.01%. 129 The number of respondents met an acceptable sample size for analysis based on the G*Power test. The sample was composed of 524 males (87.8%) and 73 females (12.2%).") All results should be in the results section of the manuscript.

RESPONSE: Thank you for your point. I moved the portion to the results section. It reads better now.

In lines 208-215,

“This study collected 597 responses representing a response rate of 23.01%. A G*Power test was conducted to draw a reliable number of respondents with effect size (.15), a Bonferroni-corrected α error probability of .05, and a .80 level of power (1-β) [35]. The result yielded a minimum sample size of 250 for valid results. The number of respondents met an acceptable sample size for analysis based on the G*Power test. The sample was composed of 524 males (87.8%) and 73 females (12.2%). The gender distribution is analogous to that of the national data of South Korean emergency workers. According to the national data in 2020, the ratio of male emergency workers to female was 9:1 [36]. The demographic information is described in Table 2.”

  1. The response rate is low (< 25%), and there are no policemen in the "emergency workers" sample. This limits the interpretation and the generalizability of the results. Inclusion and exclusion criteria are not stated, which is problematic as including "Administration" and "Training" personnel in the analysis is debatable.

RESPONSE: I included the admin and training personnel in the sample because they are part of the emergency worker population in South Korea as well. In this study, I only accessed fire departments in which rescuers, firefighters, paramedics, and administrators. I agree that this is a clear limitation of the study. I mentioned the limitations in the study limitation section. If you want me to remove the two positions from the sample and reanalyze the data in the next revision, I will definitely do it. Thank you for your understanding and please let me know. 

In lines 117-122,

“The sample respondents were emergency workers in the fire department stations in one of the major cities in South Korea. Data were collected from firefighters, paramedics, rescuers, administrators, and trainers in the fire department stations in Daegu, South Korea. In addition to firefighters, paramedics, and rescuers, administrators and trainers in the fire department stations were also included in the sample, because they deal with emergency-related tasks daily.”

In lines 368-373,

“Further, the current study included administration and training personnel in the sample. The small number of the administration and training personnel in the sample might not significantly affect the results of the analysis (n = 40, 6.7%). However, future research should focus on firefighters, paramedics, or police officers respectively to test each occupational group’s wellbeing. Third, the response rate (23.01%) was relatively low. Future research can obtain a higher response rate by promoting the benefits of participation.”

Minor comments:
6. South Korean experts in 31 emergency medical services (EMS) argue that the country needs to improve its EMS. In this sentence, the last EMS means "emergency medical system" rather than "emergency medical services" and the same abreviation should not be used.

RESPONSE: I corrected the confusion by providing the full name in the text.

In lines 35-37,

“South Korean experts in emergency medical services (EMS) argue that the country needs to improve its emergency medical system, decrease the heavy workload of emergency workers, and address the shortage of personnel [3].”

  1. "The PERMA model is also used to explain quality of life, 45 life satisfaction, and sleep quality as consequences among those with difficulties in life (e.g., 46 individuals with disabilities and emotional laborers)" - the reference is only about "Assessing College Life Adjustment of Students With Disabilities". The sentence should be updated to be more consistent with the reference cited (or a relevant reference could be added).

RESPONSE:  I corrected the point in the revision. I streamlined the sentence with the reference cited.

In lines 52-56,

“The PERMA model was used to explain life satisfaction as a consequence among those with difficulties in life (e.g., individuals with disabilities) [9]. Psychological precedents (e.g., PTSD, burnout, and depression) and PERMA were significant factors to be associated with one another [8]. The PERMA model has also been employed to account for quality of life, life satisfaction, and sleep quality [9].”

  1. Between lines 56 and 64, there is no real "flow" and the sentences do not seem to be linked together

RESPONSE: I agree with you. I delineated the paragraph with the relevant points needed to explain the model. I also added subheadings to indicate each section’s discussion on the topic for clarity and readability.

In lines 52-64,

“The PERMA model was used to explain life satisfaction as a consequence among those with difficulties in life (e.g., individuals with disabilities) [9]. Psychological precedents (e.g., PTSD, burnout, and depression) and PERMA were significant factors to be associated with one another [8]. The PERMA model has also been employed to account for quality of life, life satisfaction, and sleep quality [9].

Grounded in the PERMA model, the current study examines the precedents (PTSD, burnout, and depression) and consequences (quality of life, life satisfaction, and sleep quality) of PERMA among South Korean emergency workers. This study investigates whether both inner and outer psychological and physical factors emergency workers experience during work account for wellbeing consequences. The results of the PERMA analysis may offer new explanatory mechanisms that can contribute to improving South Korean emergency workers’ lives and developing tailored programs. Moreover, the data collected during the coronavirus disease 19 (COVID-19) may suggest emergency workers’ wellbeing management strategies during a pandemic.”

  1. Some references are incorrectly formatted ("a .80 level of power (1-*beta*) (Cohen, 1988).")

RESPONSE: I apologize for the error. I corrected the reference in the revision.

In lines 208-210,

“This study collected 597 responses representing a response rate of 23.01%. A G*Power test was conducted to draw a reliable number of respondents with effect size (.15), a Bonferroni-corrected α error probability of .05, and a .80 level of power (1-β) [35].”

Thank you, Reviewer for your constructive comments and suggestions. With the changes applied to the revision, I realize that the paper is much improved. Hope this revision meets your expectations. If I misunderstood your points, please let me know. I will correct them as you suggest.  

Reviewer 2 Report

Thank you for the opportunity to review the manuscript entitled:

 Mental and Physical Factors Influencing Wellbeing among South Korean Emergency Workers.

Before publication the work requires thorough improvement:

Typically, abbreviations are not used in abstracts.

The aim of the work requires improvement, it is not clearly specified and is not consistent with the topic. What's more in the abstract is different than at the end of the introduction.

Introduction is very chaotic and difficult to follow adopted concept, there is no place to present the results, only to describe the subject of the issue, taking into account other works in this field.

The methodology of the conducted research requires thorough improvement.

Line 107: “Data were collected from a sample of firefighters, paramedics, rescuers, and related employees (administration and training) – please explain who exactly was involved in this study?

No data regarding the course of conducted research

The author has not precisely presented the course of the research.

  • No consent from the Bioethics Committee - please provide the consent number.

You mentioned about “ I_R_B_ _of_ _t_h_e_ _r_e s_e_a_r_c_h_e_r_’s_ _u_n_i_v_e_r_s_i_t_y”_ _ - please explain the abbreviation and add the number of consent.

  • How was the recruitment of the test group looked like?
  • Please enter your inclusion and exclusion criteria
  • Please write exactly on what the sample was based on
  • Please describe the method of sampling and research facilities - some sampling algorithm?

Table No. 1 - these are the results, as a characteristic of the study group, not a methodology.

No explanation as to whether the scales used in the study were adapted to your country's conditions or only translated, if so, were back translated, was an experimental panel carried out, were the scales validated?

  • whether consent for their use has been obtained?

What scales were used in the study, is the quality of life a WHO scale /, has the author obtained consent?

The satisfaction of life – is it the SWLS scale?

There are too many unanswered questions, and this greatly undermines the validity of the research being conducted.

The results are not well presented

The results do not show the level of burnout, life satisfaction and quality of life in the study group.

It is worth calculating the regression to show which factors are key in the conducted study.

The discussion repeats the content of the introduction.

The study has numerous limitations, it should be supplemented

Throughout the work, many abbreviations are not explained, tables require a detailed description of all abbreviations.

No information about the availability of data, please complete.

Author Response

Typically, abbreviations are not used in abstracts.

RESPONSE: Thank you Reviewer for your comments. As PERMA was a key variable of the study, I could not avoid using it to explain the results in the abstract because PERMA’s full name was too long. I used the full name of PTSD in the abstract. I provided the full name of the abbreviations when it appeared first. Thank you for your understanding.

In lines 9-25,

“Abstract: Emergency workers are frequently exposed to hazardous situations and such life patterns can influence their wellbeing. This study examined the relationships among South Korean emergency workers’ precedents and consequences of positive emotion, engagement, relationship, meaning, and achievement (PERMA), a wellbeing concept, and offered solutions. A total of 597 emergency workers in Daegu, South Korea participated in a survey. This study measured post-traumatic stress disorder syndrome, burnout, depression, PERMA, quality of life, life satisfaction, and sleep quality to test the relationships. Results demonstrated that post-traumatic stress disorder syndrome and burnout predicted distracting sleep behavior and sleep health. Depression was significantly related to PERMA. The better the emergency workers’ PERMA was, the better their quality of life and life satisfaction were. PERMA significantly predicted sleep behavior, a portion of sleep quality. Depression had an indirect influence on quality of life mediated by PERMA. Post-traumatic stress disorder syndrome, burnout, and PERMA were significant predictors of low sleep health and sleep behavior. The results indicate that South Korean emergency workers struggle with depression and sleep quality. As the data were collected during the coronavirus disease 19 pandemic, individual efforts and relevant programs to improve South Korean emergency workers’ PERMA and sleep quality in a crisis are recommended. Possible solutions to improve the wellbeing of South Korean emergency workers are suggested.”

The aim of the work requires improvement, it is not clearly specified and is not consistent with the topic. What's more in the abstract is different than at the end of the introduction.

RESPONSE: I agree with you. I streamlined my research purposes in the abstract, introduction, and discussion.

In lines 57-59,

“Grounded in the PERMA model, the current study examines the precedents (PTSD, burnout, and depression) and consequences (quality of life, life satisfaction, and sleep quality) of PERMA among South Korean emergency workers.”

Introduction is very chaotic and difficult to follow adopted concept, there is no place to present the results, only to describe the subject of the issue, taking into account other works in this field.

RESPONSE I agree with you. I added subheadings to clarify the typology of the approach. I also included only related issues and reviews to the topic. Some research results were moved to the discussion section. I hope the subheadings and refined revision help you read the sections better.

In lines 30-103,

“As emergency workers (e.g., paramedics and firefighters) are frequently exposed to perilous situations resulting from car accidents, fires, and massive disasters, they tend to suffer from psychological disorder such as post-traumatic stress disorder syndrome (PTSD), burnout, and depression [1]. The emergency system in South Korea, an Asian country, is in transition. South Korea is still above the Organization for Economic Cooperation and Development (OECD) average in the annual number of deaths resulting from disasters [2]. South Korean experts in emergency medical services (EMS) argue that the country needs to improve its emergency medical system, decrease the heavy workload of emergency workers, and address the shortage of personnel [3]. For the last two decades, the South Korean EMS field has received substantial research attention in finding solutions for employees’ excessive workload and psychological health issues. Studies were conducted on the relationship between work stressors and quality of life [4], the influence of violent experiences and stressors on work satisfaction [5], and the association between violent experiences and work stress [6]. Despite such research attempts in South Korea, the studies dealt with individual factors for respective effects rather than compounding relationships. Little research has been conducted on explaining integrative mental and physical factors for the wellbeing of emergency workers in a South Korean context.

A positive psychology model offers a theoretical frame that considers multiple elements of wellbeing. Seligman recounts psychosocial wellbeing as positive emotion, engagement, relationship, meaning, and achievement (PERMA) [7]. The PERMA model regards five elements (positive emotion, engagement, relationship, meaning, and achievement) as motivators that account for wellbeing in life [7]. The PERMA model explains that the five elements can lead to a life of fulfillment and satisfaction [7]. Past research using the PERMA model found that high PERMA is negatively associated with PTSD, burnout, and depression [8]. The PERMA model was used to explain life satisfaction as a consequence among those with difficulties in life (e.g., individuals with disabilities) [9]. Psychological precedents (e.g., PTSD, burnout, and depression) and PERMA were significant factors to be associated with one another [8]. The PERMA model has also been employed to account for quality of life, life satisfaction, and sleep quality [9].

Grounded in the PERMA model, the current study examines the precedents (PTSD, burnout, and depression) and consequences (quality of life, life satisfaction, and sleep quality) of PERMA among South Korean emergency workers. This study investigates whether both inner and outer psychological and physical factors emergency workers experience during work account for wellbeing consequences. The results of the PERMA analysis may offer new explanatory mechanisms that can contribute to improving South Korean emergency workers’ lives and developing tailored programs. Moreover, the data collected during the coronavirus disease 19 (COVID-19) may suggest emergency workers’ wellbeing management strategies during a pandemic.

  • Precedents of positive emotion, engagement, relationship, meaning, and achievement (PERMA)

As a precedent to PERMA, PTSD is defined as a mental disorder stemming from experiencing terrifying events [10]. Emergency workers’ generativity, meaning in life, self-esteem, and social support are contributors to lower levels of PTSD [11]. Emergency workers with high PTSD experience low quality of life, excessive anxiety, and frights that negatively affect job implementation and family relationships [12]. The emergency workers who rescued victims in the aftermath of the 9-11 World Trade Center attacks later demonstrated high anxiety and low quality of life [13]. As seen in the research findings, PTSD likely predicts PERMA elements.

Burnout is a syndrome of physical, emotional, and mental states resulting from chronic workplace exposure that demands elevated emotion [14]. Emergency workers tend to experience burnout due to sustaining work stress [15]. As the level of burnout increases, quality of life decreases, and the health of emergency workers is affected negatively [16]. Given the research evidence, burnout would contribute to explaining PERMA.

Depression refers to a mental health state characterized by a persistently depressed mood, concern, feeling of loss, loss of self-esteem, or loss of interest in activities due to negative thoughts, one’s perceptions, environments, and future outcomes [17]. The PERMA elements are factors that negatively predict depression, stress, and anxiety [8]. Therefore, it is predicted that depression can be an influencing factor for PERMA.

  • Consequences of positive emotion, engagement, relationship, meaning, and achievement (PERMA)

The World Health Organization (WHO) defines quality of life as the perceptions about an individual’s status in life regarding the person’s goals, expectations, norms, and interests in the context of culture and value systems [18]. Emergency workers’ quality of life is influenced not only by health but also by PTSD, familial support, and health-related work restrictions [19].

Life satisfaction has been used as an indicator that tests the level of psychological wellbeing and health in many countries [20]. Butler and Kern discuss that PERMA is a substantial component to account for life satisfaction [21]. Further, positive relational quality at home and work can diminish stress and facilitate quality of life and life satisfaction [22].

Sleep is one of the crucial parameters that affects a person’s quality of life and health [23]. Due to the work characteristics of frequent shifts, emergency workers experience sleep shortage, alcohol consumption, depression, anxiety, and distressing events [24]. Research evidence suggests that emergency workers’ wellbeing is likely significantly related to sleep quality.

Based on the review, this study tests whether a group of South Korean emergency workers’ precedents (PTSD, burnout, and depression) predict PERMA. Another set of predictions is to examine whether PERMA predicts consequences (quality of life, life satisfaction, and sleep quality). A proposed model is displayed in Figure 1.”

The methodology of the conducted research requires thorough improvement.

RESPONSE: I fine-tuned the section and added necessary information for clarification. As you pointed out, I detailed sample information, data collection procedures, research design, sampling, and measurement.

In lines 112-202,

“This study used a purposive sampling method to collect data from emergency workers in South Korea. The goal of a purposive sample is to collect data from a representative sample of the population. This study attempted to accomplish the goal by applying expert knowledge to the population to select a nonrandom sample that represents a cross-section of the population [25]. The study’s population entailed all employees of the fire department stations in South Korea. This study used cross-sectional data collected from a survey of emergency workers. The sample respondents were emergency workers in the fire department stations in one of the major cities in South Korea. Data were collected from firefighters, paramedics, rescuers, administrators, and trainers in the fire department stations in Daegu, South Korea. In addition to firefighters, paramedics, and rescuers, administrators and trainers in the fire department stations were also included in the sample, because they deal with emergency-related tasks daily. Daegu was chosen for the following two reasons. First, the city is the third largest in South Korea and has a population of 2.5 million as of 2019. Second, as a metropolitan city, Daegu is the site of the subway arson disaster of 1995, the propane gas tank explosion of 1995, and the largest number of COVID-19 victims in February-March 2020.

A total of nine fire department stations (Regional Headquarter, Central, East, West, North, Suseong, Dalseo, Dalseong, and Kangseo) cover the districts in Daegu. Through the sampling frame of the emergency workers of the fire department stations in Daegu, the researcher sent a survey participation request to six stations (Headquarter, North, East, Central, Suseong, and Dalseo) and asked a representative of each station to distribute an online survey link to all emergency workers via email. Using purposive sampling, the six stations were chosen based on the researcher’s professional networks. In the requests, the researcher also asked the representatives to share the survey link with the other three stations (West, Dalseong, and Kanseo). As a result, 2,595 firefighters, paramedics, and related employees (Regional Headquarter = 216, Central = 335, East = 313, West = 364, North = 302, Suseong = 318, Dalseo =232, Dalseong = 248, and Kangseo = 267) were reached for survey participation.

Upon approval from the Institutional Review Board (IRB) of the researcher’s university (Ethical Code: SM-202005-035-2), a survey was conducted from July 1 – September 8, 2020, in which participants were provided an online survey link. The respondents who received a survey link via email visited the survey site and were asked to consent to participate in the survey. Only those who consented participated in the survey. The questionnaire was fully anonymous. No private information was collected. The questionnaire adapted the original scales developed in past studies and was translated into Korean. To verify accuracy and objectivity, a third researcher in a different field reviewed both the English and Korean versions. The third researcher verified the translation. Finally, the IRB confirmed the appropriateness of the questionnaire’s wording and approved translation. 

2.2. Measurement

The current study used a total of seven variables as measurement instruments. An exploratory factor analysis (EFA) was conducted for the instruments’ validity and reliability, because the scales were reconstructed for the purpose of this study. Each component was determined based on eigenvalues over 1.0 and coefficients over .5 [26]. Component reliabilities were measured with Cronbach’s α for reliability with over .7 as an acceptable scale [27]. The EFA used the Principal Component with Varimax Rotation method. The EFA drew reliable components from eight items of PTSD, 12 items of burnout, eight items of PERMA, four items of quality of life, five items of life satisfaction, and 11 items of sleep quality. This study also conducted scale validity tests with the Kaiser-Meyer-Olkin (KMO) measure of sampling adequacy and Bartlett’s test of specificity. A single latent variable was created for each scale by summing and averaging. The latent variables were used for correlation and regression analysis in the preliminary phase. Factor coefficients, eigenvalues, variance explained, KMO measure of sampling adequacy, Bartlett’s test of specificity for model fit, and Cronbach’s Alpha were presented in Table 1. All factor Items are listed in the Appendix.

2.2.1. PTSD

PTSD measured the level of disturbing memories and thoughts due to stressful experiences of the past [28] and was rephrased for the current study’s context. Due to redundancies among the items in the other measurements of this study, only mutually exclusive eight items were selected (from 1 = Not at all to 5 = Very much so). An EFA yielded one factor for the eight items.

2.2.2. Burnout

The items of burnout measured the degree to which the respondents felt emotional fatigue in varying forms [29] (from 1 = Not at all to 5 = Very much so). An EFA of 12 items produced unidimensionality with one factor. There were no item dropouts.

2.2.3. Depression

Depression was assessed with the state of emotional anxiety and stress [30]. Four items with high coefficients verified in past research were selected to measure (from 1 = Do not apply to me to 5 = Applies to me all the time). An EFA found one dimension without any item dropouts.

2.2.4. PERMA

As the wellbeing measure, a total of eight PERMA items were adapted for an EFA [31] (from 1 = Not at all to 5 = Always) and rephrased for the South Korean emergency worker context. After removing overlapping items with the other measurement scales, eight items were chosen for analysis. The eight items represented the PERMA elements, respectively. The scale assessed the levels of PERMA. Different from the anticipation of multiple factors, an EFA generated a single scale for the eight items.

2.2.5. Quality of life

The quality of life measure was operationally defined as the level of quality the respondents feel in the health, joy, and meaning of daily life [32] (from 1 = Not at all to 5 = Very much so). This study adapted the World Health Organization Quality of Life-brief (WHOQOL-BREF) scale and selected items with top four factor coefficients in past research [31]. The result of an EFA with four items produced unidimensionality.

2.2.6. Life satisfaction

The scale assessed the degree of satisfactory feelings in everyday life and used the satisfaction with life scale (SWLS) [33] (from 1 = Not at all to 5 = Very much so). Only five items with top factor coefficients in past research were adapted and analyzed in EFA [32]. The analysis yielded one factor without any dropouts.

2.2.7. Sleep quality

The 11 items of sleep quality measured health and behaviors during sleep [34] (from 1 = Not at all to 5 = Very much so). High scores indicated low sleep quality. An EFA yielded two components: sleep health (seven items) and sleep behavior (four items).

2.2.8. Statistical analysis

This study used Statistical Product and Service Solutions (SPSS) 26.0 for demographic characteristics, exploratory factor analysis, correlations, and regressions. A structural equation modeling test was implemented with Analysis of a Moment Structures (AMOS) 26.0.”

Line 107: “Data were collected from a sample of firefighters, paramedics, rescuers, and related employees (administration and training) – please explain who exactly was involved in this study?

RESPONSE: All of them because they are the members of the unit. I elaborated on sample information in the section. All of them were included in analysis because they are the members of an emergency worker unit. In South Korea, fire departments encompass the emergency workers altogether. In addition to firefighters, paramedics, and rescuers, administrators and trainers were also included in analysis because they are part of the unit. In a One-Way ANOVA analysis of the dependent variables by position, there were no significant differences in their levels of PTSD, burnout, depression, PERMA, life satisfaction, quality of life, and sleep quality. Therefore, administrators and trainers’ level of physical and psychological issues were identical.

In lines 117-122,

“The sample respondents were emergency workers in the fire department stations in one of the major cities in South Korea. Data were collected from firefighters, paramedics, rescuers, administrators, and trainers in the fire department stations in Daegu, South Korea. In addition to firefighters, paramedics, and rescuers, administrators and trainers in the fire department stations were also included in the sample, because they deal with emergency-related tasks daily.”

In lines 215-221,

“For the comparison purpose of the variables, a One-Way Analysis of Variance (ANOVA) analysis was used for the dependent variables by position. Results demonstrated that there were no significant differences in PTSD, burnout, depression, PERMA, quality of life, life satisfaction, sleep health, and sleep behavior by position. The results suggest that regardless of positions, the sampled emergency workers – including rescuers, paramedics, firefighters, administrators, and trainers – showed the identical levels on the dependent variables (see Table 3 for Means of Variables).”

No data regarding the course of conducted research. The author has not precisely presented the course of the research.

RESPONSE: I elaborated on the procedures of research design in the method section.

In lines 112-146,

“This study used a purposive sampling method to collect data from emergency workers in South Korea. The goal of a purposive sample is to collect data from a representative sample of the population. This study attempted to accomplish the goal by applying expert knowledge to the population to select a nonrandom sample that represents a cross-section of the population [25]. The study’s population entailed all employees of the fire department stations in South Korea. This study used cross-sectional data collected from a survey of emergency workers. The sample respondents were emergency workers in the fire department stations in one of the major cities in South Korea. Data were collected from firefighters, paramedics, rescuers, administrators, and trainers in the fire department stations in Daegu, South Korea. In addition to firefighters, paramedics, and rescuers, administrators and trainers in the fire department stations were also included in the sample, because they deal with emergency-related tasks daily. Daegu was chosen for the following two reasons. First, the city is the third largest in South Korea and has a population of 2.5 million as of 2019. Second, as a metropolitan city, Daegu is the site of the subway arson disaster of 1995, the propane gas tank explosion of 1995, and the largest number of COVID-19 victims in February-March 2020.

A total of nine fire department stations (Regional Headquarter, Central, East, West, North, Suseong, Dalseo, Dalseong, and Kangseo) cover the districts in Daegu. Through the sampling frame of the emergency workers of the fire department stations in Daegu, the researcher sent a survey participation request to six stations (Headquarter, North, East, Central, Suseong, and Dalseo) and asked a representative of each station to distribute an online survey link to all emergency workers via email. Using purposive sampling, the six stations were chosen based on the researcher’s professional networks. In the requests, the researcher also asked the representatives to share the survey link with the other three stations (West, Dalseong, and Kanseo). As a result, 2,595 firefighters, paramedics, and related employees (Regional Headquarter = 216, Central = 335, East = 313, West = 364, North = 302, Suseong = 318, Dalseo =232, Dalseong = 248, and Kangseo = 267) were reached for survey participation.

Upon approval from the Institutional Review Board (IRB) of the researcher’s university (Ethical Code: SM-202005-035-2), a survey was conducted from July 1 – September 8, 2020, in which participants were provided an online survey link. The respondents who received a survey link via email visited the survey site and were asked to consent to participate in the survey. Only those who consented participated in the survey. The questionnaire was fully anonymous. No private information was collected. The questionnaire adapted the original scales developed in past studies and was translated into Korean. To verify accuracy and objectivity, a third researcher in a different field reviewed both the English and Korean versions. The third researcher verified the translation. Finally, the IRB confirmed the appropriateness of the questionnaire’s wording and approved translation.”

  • No consent from the Bioethics Committee - please provide the consent number.

Yes, I provided the ethical code in the revision.

In lines 137-139,

“Upon approval from the Institutional Review Board (IRB) of the researcher’s university (Ethical Code: SM-202005-035-2), a survey was conducted from July 1 – September 8, 2020, in which participants were provided an online survey link.”

You mentioned about “ I_R_B_ _of_ _t_h_e_ _r_e s_e_a_r_c_h_e_r_’s_ _u_n_i_v_e_r_s_i_t_y”_ _ - please explain the abbreviation and add the number of consent.

RESPONSE:  I provided the full name of the abbreviation in the revision.

In lines 137-139,

“Upon approval from the Institutional Review Board (IRB) of the researcher’s university (Ethical Code: SM-202005-035-2), a survey was conducted from July 1 – September 8, 2020, in which participants were provided an online survey link.”

  • How was the recruitment of the test group looked like?

RESPONSE: I mentioned and detailed the recruitment process in the revision.

In lines 112-136,

“This study used a purposive sampling method to collect data from emergency workers in South Korea. The goal of a purposive sample is to collect data from a representative sample of the population. This study attempted to accomplish the goal by applying expert knowledge to the population to select a nonrandom sample that represents a cross-section of the population [25]. The study’s population entailed all employees of the fire department stations in South Korea. This study used cross-sectional data collected from a survey of emergency workers. The sample respondents were emergency workers in the fire department stations in one of the major cities in South Korea. Data were collected from firefighters, paramedics, rescuers, administrators, and trainers in the fire department stations in Daegu, South Korea. In addition to firefighters, paramedics, and rescuers, administrators and trainers in the fire department stations were also included in the sample, because they deal with emergency-related tasks daily. Daegu was chosen for the following two reasons. First, the city is the third largest in South Korea and has a population of 2.5 million as of 2019. Second, as a metropolitan city, Daegu is the site of the subway arson disaster of 1995, the propane gas tank explosion of 1995, and the largest number of COVID-19 victims in February-March 2020.

A total of nine fire department stations (Regional Headquarter, Central, East, West, North, Suseong, Dalseo, Dalseong, and Kangseo) cover the districts in Daegu. Through the sampling frame of the emergency workers of the fire department stations in Daegu, the researcher sent a survey participation request to six stations (Headquarter, North, East, Central, Suseong, and Dalseo) and asked a representative of each station to distribute an online survey link to all emergency workers via email. Using purposive sampling, the six stations were chosen based on the researcher’s professional networks. In the requests, the researcher also asked the representatives to share the survey link with the other three stations (West, Dalseong, and Kanseo). As a result, 2,595 firefighters, paramedics, and related employees (Regional Headquarter = 216, Central = 335, East = 313, West = 364, North = 302, Suseong = 318, Dalseo =232, Dalseong = 248, and Kangseo = 267) were reached for survey participation.”

  • Please enter your inclusion and exclusion criteria

RESPONSE: I included all respondents from the fire department stations for several reasons. All of them (rescuers, paramedics, firefighters, administrators, and trainers) are the employees of the stations. They take part in each portion of the duty required in the stations. As seen in the comparison of the dependent variables by position, there were no significant differences indicating that all positions feel identical in the precedents and consequences of PERMA, a wellbeing concept.

In 112-122,

“This study used a purposive sampling method to collect data from emergency workers in South Korea. The goal of a purposive sample is to collect data from a representative sample of the population. This study attempted to accomplish the goal by applying expert knowledge to the population to select a nonrandom sample that represents a cross-section of the population [25]. The study’s population entailed all employees of the fire department stations in South Korea. This study used cross-sectional data collected from a survey of emergency workers. The sample respondents were emergency workers in the fire department stations in one of the major cities in South Korea. Data were collected from firefighters, paramedics, rescuers, administrators, and trainers in the fire department stations in Daegu, South Korea. In addition to firefighters, paramedics, and rescuers, administrators and trainers in the fire department stations were also included in the sample, because they deal with emergency-related tasks daily.”

  • Please write exactly on what the sample was based on

RESPONSE: Fire department station employees in South Korea were the population of the study. In Korea, emergency workers in fire department stations consist of rescuers, paramedics, firefighters, administrators, and trainers.

In lines 115-122,

“The study’s population entailed all employees of the fire department stations in South Korea. This study used cross-sectional data collected from a survey of emergency workers. The sample respondents were emergency workers in the fire department stations in one of the major cities in South Korea. Data were collected from firefighters, paramedics, rescuers, administrators, and trainers in the fire department stations in Daegu, South Korea. In addition to firefighters, paramedics, and rescuers, administrators and trainers in the fire department stations were also included in the sample, because they deal with emergency-related tasks daily.”

  • Please describe the method of sampling and research facilities - some sampling algorithm?

RESPONSE: I used a purposive nonrandom sampling method. I accessed the respondents through email using my professional networks. It was an online survey. I collected the data from a survey webpage and analyzed in SPSS and AMOS.

In lines 112-115,

“The study’s population entailed all employees of the fire department stations in South Korea. This study used cross-sectional data collected from a survey of emergency workers. The sample respondents were emergency workers in the fire department stations in one of the major cities in South Korea. Data were collected from firefighters, paramedics, rescuers, administrators, and trainers in the fire department stations in Daegu, South Korea. In addition to firefighters, paramedics, and rescuers, administrators and trainers in the fire department stations were also included in the sample, because they deal with emergency-related tasks daily.” 

Table No. 1 - these are the results, as a characteristic of the study group, not a methodology.

RESPONSE: Thanks for your point. I moved the portion to the results section.

In lines 208-215,

“This study collected 597 responses representing a response rate of 23.01%. A G*Power test was conducted to draw a reliable number of respondents with effect size (.15), a Bonferroni-corrected α error probability of .05, and a .80 level of power (1-β) [35].  The result yielded a minimum sample size of 250 for valid results. The number of respondents met an acceptable sample size for analysis based on the G*Power test. The sample was composed of 524 males (87.8%) and 73 females (12.2%). The gender distribution is analogous to that of the national data of South Korean emergency workers. According to the national data in 2020, the ratio of male emergency workers to female was 9:1 [36]. The demographic information is described in Table 2.”

No explanation as to whether the scales used in the study were adapted to your country's conditions or only translated, if so, were back translated, was an experimental panel carried out, were the scales validated?

RESPONSE: Yes, the translation was verified by the IRB. The scales were all validated by previous research that used the scale. As you can see, the reliability (Cronbach’s alphas) and validity (KMO measure of sampling adequacy and Bartlett’s test of specificity by Chi-square) were provided. The questionnaire was translated into Korean and it fit for the Korean emergency workers’ context. The translation was reviewed, verified, and approved.

In lines 148-160,

“This study collected 597 responses representing a response rate of 23.01%. A G*Power test was conducted to draw a reliable number of respondents with effect size (.15), a Bonferroni-corrected α error probability of .05, and a .80 level of power (1-β) [35].  The result yielded a minimum sample size of 250 for valid results. The number of respondents met an acceptable sample size for analysis based on the G*Power test. The sample was composed of 524 males (87.8%) and 73 females (12.2%). The gender distribution is analogous to that of the national data of South Korean emergency workers. According to the national data in 2020, the ratio of male emergency workers to female was 9:1 [36]. The demographic information is described in Table 2.”

In lines 139-146,

“The respondents who received a survey link via email visited the survey site and were asked to consent to participate in the survey. Only those who consented participated in the survey. The questionnaire was fully anonymous. No private information was collected. The questionnaire adapted the original scales developed in past studies and was translated into Korean. To verify accuracy and objectivity, a third researcher in a different field reviewed both the English and Korean versions. The third researcher verified the translation. Finally, the IRB confirmed the appropriateness of the questionnaire’s wording and approved translation.”

  • whether consent for their use has been obtained?

RESPONSE: Yes, all the scales were already available and used in past research. In the case of the WHOQOL scale, its scale items are open to the public for use on the WHO website.

What scales were used in the study, is the quality of life a WHO scale /, has the author obtained consent?

RESPONSE: The WHOQOL scale is open to the public to use. The scale was also used and verified in the cited past research.

In lines 185-189,

“The respondents who received a survey link via email visited the survey site and were asked to consent to participate in the survey. Only those who consented participated in the survey. The questionnaire was fully anonymous. No private information was collected. The questionnaire adapted the original scales developed in past studies and was translated into Korean. To verify accuracy and objectivity, a third researcher in a different field reviewed both the English and Korean versions. The third researcher verified the translation. Finally, the IRB confirmed the appropriateness of the questionnaire’s wording and approved translation.”

The satisfaction of life – is it the SWLS scale?

RESPONSE: Yes, I used the SWLS scale for this study.

In lines 191-194,

“The scale assessed the degree of satisfactory feelings in everyday life and used the satisfaction with life scale (SWLS) [33] (from 1 = Not at all to 5 = Very much so). Only five items with top factor coefficients in past research were adapted and analyzed in EFA [32]. The analysis yielded one factor without any dropouts.”

There are too many unanswered questions, and this greatly undermines the validity of the research being conducted.

RESPONSE: Thank you for your questions and points. I tried to answer them all as possible as I could. I added all possible information to verify the validity of the research. If my answers need to address further, please let me know.

The results are not well presented. The results do not show the level of burnout, life satisfaction and quality of life in the study group.

RESPONSE: I presented the descriptive data results of the variables. In both regression and SEM, the variables were tested. I tried my best to address your points in the results section.

It is worth calculating the regression to show which factors are key in the conducted study.

RESPONSE: Yes, as a preliminary analysis before SEM, I provided the results of regression which turned out to be identical with the SEM results except few relationships. I included the results to help view the structure of the data.

In lines 236-248,

“As an additional preliminary analysis, this study conducted multiple regressions with the same independent and dependent variables. In predicting PERMA after controlling for demographics, burnout (β = -.17, p < .05) and depression (β = -.36, p < .001) were negatively associated with PERMA. The regression model explained 25.8% of the total variance (F = 22.73, p < .001). Another regression analysis with PTSD, burnout, depression, and PERMA as the independent variables and quality of life, life satisfaction, and sleep quality as the dependent variables was implemented. PERMA significantly predicted quality of life (β = .62, p < .001), life satisfaction (β = .69, p < .001), sleep health (β = -.09, p < .01), and sleep behavior (β = -.09, p < .05). PTSD predicted sleep health (β = .29, p < .001) and sleep behavior (β = .25, p < .001). Burnout predicted quality of life (β = -.13, p < .05), life satisfaction (β = -.29, p < .001), sleep health (β = .24, p < .001), and sleep behavior (β = .40, p < .01). Depression predicted sleep health (β = .18, p < .01). The regression models explained 49.4% of the total variance for quality of life (F = 57.14, p < .001), 64.8% for life satisfaction (F = 107.84, p < .001), 51.4% for sleep health (F = 61.97, p < .001), and 35.20% for sleep behavior (F = 31.90, p < .001).”

The discussion repeats the content of the introduction.

RESPONSE: I revised and mentioned in the following format: result summary, implication, and suggestion.

In lines 287-356,

“The present study examined the precedents and consequences of PERMA, a wellbeing concept, among emergency workers in South Korea. Key findings suggest that a sample of South Korean emergency workers experience depression, which negatively influences PERMA. Moreover, PERMA is a significant indicator in predicting the respondents’ quality of life and life satisfaction. Particularly, PTSD and burnout are deterrents of sleep quality. The results suggest that the respondents tend to grapple with depression, sleep health, and sleep behavior.

The result of the relationship between PTSD and PERMA is consistent with past research that found a significant relationship between PTSD and heavy workload but not PERMA [38]. The result indicates that the respondents may perceive the compounding influence of PTSD at the work level rather than solely at the personal level. Meanwhile, PTSD significantly predicted negative sleep behavior. The result is congruent with past studies [39, 40], which found that sleep difficulty was deteriorated by PTSD and emphasized the importance of sleep quality for the improvement of PTSD. Sleep might be an issue of attention for South Korean emergency workers. The top concern of South Korean emergency workers was sleep disturbance, including insomnia and sleep apnea [41]. Experts on the treatment of emergency workers’ PTSD suggest working out, staying connected to support their fellow emergency workers, and getting educated about PTSD as solutions [42]. With these efforts, emergency workers may also improve sleep quality. Specific and practical therapies based on the results may improve the sleep quality of employees in the EMS field [43].

The nonsignificant relationship between burnout and PERMA is not consistent with past research [44]. Perhaps the relationship can be better explained with a moderator such as resilience [45]. Meanwhile, burnout negatively explained life satisfaction. Since burnout hinders a satisfying life, emergency personnel may seek some ideas of work-life balance [46]. As burnout creates a chronic motivational challenge, coping mechanisms such as trying to live an energetic life, developing positive aspects of life, and setting top priorities in daily lives can contribute to quality of life [47]. Those with PERMA in life could experience burnout less than those without PERMA [48]. Future research needs to elaborate on multiple factors including workload and emotional labor [49] in emergency workers’ work environment to account for burnout and PERMA.

The significant relationship between depression and PERMA implies that the depression of emergency workers can be a deterrent for life engagement, goal attainment, accountability, and happiness. In the SEM analysis, depression significantly predicted quality of life but became nonsignificant when PERMA was added as a mediator between depression and quality of life. The result suggests that PERMA may play a crucial role between emergency workers’ depression and quality of life [50]. As related research affirms [51], South Korean emergency workers’ efforts and some intervention programs and education to improve wellbeing may alleviate depression and enhance quality of life.

The positive relationship between PERMA and quality of life indicates that the respondents regard PERMA as an element for a joyful, meaningful, and healthy life with high morale [52]. Previous research that found quality of life facilitating resilience and health supports the current result [53]. Sustainable support for first responders is a significant enhancer for wellbeing and quality of life [12]. The positive relationship between PERMA and life satisfaction suggests emergency workers’ heightened attention to PERMA as a solution. Fire department stations’ concerted efforts at the organizational level to improve PERMA – which can facilitate life satisfaction, quality of life, and sleep quality – are recommended. The significant relationships among PTSD, burnout, and sleep quality imply that sleep health and sleep behavior are determined largely by the compounding factors, which suggests that increased attention is needed to facilitate better sleep quality [54]. Low PERMA could cause insomnia and shortened sleep duration [55]. The significant predictability of sleep behavior for sleep health implies that the respondents’ unhealthy sleep behavior may lead to hampered sleep wellbeing.

Through the results, the current study contributes to the EMS field both theoretically and practically. Theoretically, scant research attempts have comprehensively examined PTSD, burnout, depression, PERMA, quality of life, life satisfaction, and sleep quality among emergency workers. The current study is the first in South Korea to test the relationships between precedents and consequences of PERMA, a wellbeing concept, among emergency workers. An advanced theoretical model of PERMA may be developed from this research [7]. The theoretical framework used in this study can be applied to similar professionals, such as emotional laborers, hospital residents, and social welfare workers in both the private and public sectors. This study was conducted during the COVID-19 pandemic. Emergency workers were at the forefront of the crisis as first responders and coped with fear, anxiety, and uncertainty [56]. The significant relationships found in this study may reflect emergency workers’ work stress and consequences during a pandemic. The results demonstrate that South Korean emergency workers are distracted by sleep quality, which is affected by negative mental disorders such as PTSD and burnout amidst the COVID-19 pandemic.

Practically, the use of PERMA as a key factor delineating precedents and consequences of this study may offer ideas of program development (e.g., meaning of life, energetic life, top priorities in life, positive aspects of life, interaction with colleagues, healthy sleep, etc.) that can enhance the lives of emergency workers. The results of the present study suggest the factors (depression, PERMA, sleep quality, PTSD-sleep quality, and burnout-sleep quality) to emphasize further in program development. As such, the EMS field – including emergency workers in South Korea – may create tailored intervention programs for work-life balance. As emergency workers experience elevated mental disturbance levels during a pandemic [57], the consideration of PERMA and sleep quality may provide coping strategies for emergency workers in South Korea.”

The study has numerous limitations, it should be supplemented

RESPONSE: I revised. I added all possible limitations. I will conduct future research that can remedy the limitations of the current study.

In lines 360-380,

“This study has some conceptual and methodological limitations. First, the cross-sectional data from a city do not represent the entire population of South Korean emergency workers. Therefore, a national survey can be conducted in future research for representative results. Second, the structure of the sample had some limitations. The sample had only a small number of female emergency workers. Future research, therefore, could obtain a larger sample of female emergency worker participants by surveying medical emergency rooms or nurses. Police officers were not included in this emergency worker sample. Fire department stations and police department stations are two independent units in South Korea. The exclusion of police officers may limit the interpretation and the generalizability of the results. Further, the current study included administration and training personnel in the sample. The small number of the administration and training personnel in the sample might not significantly affect the results of the analysis (n = 40, 6.7%). However, future research should focus on firefighters, paramedics, or police officers respectively to test each occupational group’s wellbeing. Third, the response rate (23.01%) was relatively low. Future research can obtain a higher response rate by promoting the benefits of participation.  Fourth, with programs developed for emergency workers’ wellbeing, studies on the effects of the programs may be needed. Fifth, this study used some representing items of PERMA rather than the entire list of them. Had all five elements of PERMA as individual variables been used for analysis, more detailed relationships could have been yielded. Further, future research can apply the PERMA-V (vitality) model to strengthen the explanatory power of the model [58].  Finally, a comparative study of multiple countries’ emergency workers may provide cultural and international differences in the precedents and consequences of PERMA.”

Throughout the work, many abbreviations are not explained, tables require a detailed description of all abbreviations.

RESPONSE: I added full names of the abbreviations in the text and tables. If my responses do not address your abbreviation questions, please let me know.

In lines 402-405 for example,

“Abbreviations

PTSD               Post-Traumatic Stress Disorder Syndrome
PERMA            Positive emotion, Engagement, Relationship, Meaning, and Achievement
COVID-19         Coronavirus Disease 19”

No information about the availability of data, please complete.

RESPONSE: I am willing to provide the data. The data is available. I probably misread the data availability option. Thank you.

Thank you, Reviewer for your constructive comments and suggestions. With the changes applied to the revision, I realize that the paper is much improved. Hope this revision meets your expectations. If I misunderstood your points, please let me know. I will correct them as you suggest. 

Reviewer 3 Report

This is a good study. Well thought out, structured and necessary to know in the current situation.

The fact that it is dedicated to a specific population and with a small number of female sample requires that it be presented as a limitation.

However, this does not mean that the study is not important. The authors have made a good introduction. The method section is well resolved.

I only recommend that you add a short paragraph that describes the procedure and a line that indicates the type of research design.

The results are well thought out and interesting.

Conclusions and discussion. They are well done, but the authors should add the following: What limitations does your study have? One of them is the sample. What is the prospect or improvement that can be made? And finally, how does your study contribute to current practice and theory? With minor changes, the article can be published.  

Author Response

This is a good study. Well thought out, structured and necessary to know in the current situation.

The fact that it is dedicated to a specific population and with a small number of female sample requires that it be presented as a limitation.

RESPONSE: Thank you for your valuable comment. I added that information as a study limitation.

In lines 363-365,

“The sample had only a small number of female emergency workers. Future research, therefore, could obtain a larger sample of female emergency worker participants by surveying medical emergency rooms or nurses.”

However, this does not mean that the study is not important. The authors have made a good introduction. The method section is well resolved.

RESPONSE: Thank you.

I only recommend that you add a short paragraph that describes the procedure and a line that indicates the type of research design.

RESPONSE: I added detailed information about research design, data collection procedures, and the sample.

In lines 111-115,

“This study used a purposive sampling method to collect data from emergency workers in South Korea. The goal of a purposive sample is to collect data from a representative sample of the population. This study attempted to accomplish the goal by applying expert knowledge to the population to select a nonrandom sample that represents a cross-section of the population [25].” 

The results are well thought out and interesting.

Conclusions and discussion. They are well done, but the authors should add the following: What limitations does your study have? One of them is the sample. What is the prospect or improvement that can be made? And finally, how does your study contribute to current practice and theory? With minor changes, the article can be published.  

RESPONSE: Thanks for your suggestion. I added study limitation and theoretical and practical implications to the revision. I also mentioned contributions this study could make to the field.

In lines 335-356,

“Through the results, the current study contributes to the EMS field both theoretically and practically. Theoretically, scant research attempts have comprehensively examined PTSD, burnout, depression, PERMA, quality of life, life satisfaction, and sleep quality among emergency workers. The current study is the first in South Korea to test the relationships between precedents and consequences of PERMA, a wellbeing concept, among emergency workers. An advanced theoretical model of PERMA may be developed from this research [7]. The theoretical framework used in this study can be applied to similar professionals, such as emotional laborers, hospital residents, and social welfare workers in both the private and public sectors. This study was conducted during the COVID-19 pandemic. Emergency workers were at the forefront of the crisis as first responders and coped with fear, anxiety, and uncertainty [56]. The significant relationships found in this study may reflect emergency workers’ work stress and consequences during a pandemic. The results demonstrate that South Korean emergency workers are distracted by sleep quality, which is affected by negative mental disorders such as PTSD and burnout amidst the COVID-19 pandemic.

Practically, the use of PERMA as a key factor delineating precedents and consequences of this study may offer ideas of program development (e.g., meaning of life, energetic life, top priorities in life, positive aspects of life, interaction with colleagues, healthy sleep, etc.) that can enhance the lives of emergency workers. The results of the present study suggest the factors (depression, PERMA, sleep quality, PTSD-sleep quality, and burnout-sleep quality) to emphasize further in program development. As such, the EMS field – including emergency workers in South Korea – may create tailored intervention programs for work-life balance. As emergency workers experience elevated mental disturbance levels during a pandemic [57], the consideration of PERMA and sleep quality may provide coping strategies for emergency workers in South Korea.”

In lines 383-396,

“Through the results, the current study contributes to the EMS field both theoretically and practically. Theoretically, scant research attempts have comprehensively examined PTSD, burnout, depression, PERMA, quality of life, life satisfaction, and sleep quality among emergency workers. The current study is the first in South Korea to test the relationships between precedents and consequences of PERMA, a wellbeing concept, among emergency workers. An advanced theoretical model of PERMA may be developed from this research [7]. The theoretical framework used in this study can be applied to similar professionals, such as emotional laborers, hospital residents, and social welfare workers in both the private and public sectors. This study was conducted during the COVID-19 pandemic. Emergency workers were at the forefront of the crisis as first responders and coped with fear, anxiety, and uncertainty [56]. The significant relationships found in this study may reflect emergency workers’ work stress and consequences during a pandemic. The results demonstrate that South Korean emergency workers are distracted by sleep quality, which is affected by negative mental disorders such as PTSD and burnout amidst the COVID-19 pandemic.

Practically, the use of PERMA as a key factor delineating precedents and consequences of this study may offer ideas of program development (e.g., meaning of life, energetic life, top priorities in life, positive aspects of life, interaction with colleagues, healthy sleep, etc.) that can enhance the lives of emergency workers. The results of the present study suggest the factors (depression, PERMA, sleep quality, PTSD-sleep quality, and burnout-sleep quality) to emphasize further in program development. As such, the EMS field – including emergency workers in South Korea – may create tailored intervention programs for work-life balance. As emergency workers experience elevated mental disturbance levels during a pandemic [57], the consideration of PERMA and sleep quality may provide coping strategies for emergency workers in South Korea.”

Thank you, Reviewer for your constructive comments and suggestions. With the changes applied to the revision, I realize that the paper is much improved. Hope this revision meets your expectations. If I misunderstood your points, please let me know. I will correct them as you suggest. 

Reviewer 4 Report

Estimated Author,

Estimated Editors,

I've read with interest and appreciated this very high quality paper on Mental and Physical Factors Influencing Wellbeing among South Korean Emergency Workers.

In my opinion, this paper deserves a full publication as it is by now, without any further improvement.

Very nice job!

Author Response

I've read with interest and appreciated this very high quality paper on Mental and Physical Factors Influencing Wellbeing among South Korean Emergency Workers.

In my opinion, this paper deserves a full publication as it is by now, without any further improvement.

Very nice job!

RESPONSE: Thank you, Reviewer for your constructive comments and suggestions. I made some changes to the revision per the other reviewers’ comments and suggestions. I realize that the paper is much improved. Thank you again for your review.

Round 2

Reviewer 1 Report

Dear author,

Thank you for this extensive overhaul of your manuscript.

I have no further comment to make.

Reviewer 2 Report

Accept in current form, as you performed great work.